# Fast mode decomposition in few-mode fibers

Egor S. Manuylovich [1,2 ✉], Vladislav V. Dvoyrin[1,3] & Sergei K. Turitsyn [1,3]

Retrieval of the optical phase information from measurement of intensity is of a high interest because this would facilitate simple and cost-efficient techniques and devices. In scientific and industrial applications that exploit multi-mode fibers, a prior knowledge of spatial mode structure of the fiber, in principle, makes it possible to recover phases using measured intensity distribution. However, current mode decomposition algorithms based on the analysis of the intensity distribution at the output of a few-mode fiber, such as optimization methods or neural networks, still have high computational costs and high latency that is a serious impediment for applications, such as telecommunications. Speed of signal processing is one of the key challenges in this approach. We present a high-performance mode decomposition algorithm with a processing time of tens of microseconds. The proposed mathematical algorithm that does not use any machine learning techniques, is several orders of magnitude faster than the state-of-the-art deep-learning-based methods. We anticipate that our results can stimulate further research on algorithms beyond popular machine learning methods and they can lead to the development of low-cost phase retrieval receivers for various applications of few-mode fibers ranging from imaging to telecommunications.

¹ Aston Institute of Photonic Technologies, Aston University, Birmingham B4 7ET, UK. ² Kotelnikov Institute of Radioengineering and Electronics of RAS, Moscow 125009, Russia. ³ Aston-NSU Center for Photonics, Novosibirsk State University, Novosibirsk 630090, Russia. ✉email: e.manuylovich@aston.ac.uk

The resurgence of interest in multi-mode fibers and in few-mode fibers (FMFs), in particular, in telecommunications is mainly due to the recognition of the fact that only the application of parallel channels can cope with the fast-growing demand on capacity of communication systems. Spatial mode-division multiplexing is one of the actively studied approaches to provide for high capacity optical links. FMFs are widely believed to provide the optimal practical balance between the highly important possibility to increase the communication capacity compared to single-mode fibers and the growing complexity of signal processing when dealing with many transversal modes[1]. Few-mode fiber is also an attractive platform for non-telecom application fields such as imaging[2], microwave photonics[3], optical sensing[4], and fundamental studies of the complex nonlinear spatiotemporal dynamics including spatiotemporal solitons and optical beam self-cleaning[5–11]. Larger mode areas (as compared to SMF) provided by FMFs suppress nonlinear effects and improve damage threshold, making a pathway to the development of novel high-power laser systems. Control and measurement of the optical phase at the FMF output is important for many scientific and industrial applications.

Implementation of spatial division multiplexing in modern coherent communication systems that use both the amplitude and phase of the optical signal is based on the sophisticated and relatively expensive multiple-input multiple-output (MIMO) processing schemes, which rely on bulk optics[12–14]. The technical challenge in using multi-core and FMFs to increase system capacity is related to the need to use adaptive MIMO processing techniques for spatial de-multiplexing and to dynamically compensate for the differential mode group delays. The complexity of signal processing is growing quickly with the number of modes. Alternative approaches to mode de-multiplexing with a reduced complexity even for few mode fibers are of great interest for both telecom and other applications. Highly attractive solution, from a practical viewpoint, would be the phase retrieval for optical beams or pulses from intensity-only measurements[15]. Recently, fiber transmission in three-modes-both-polarization using direct detection (intensity-only measurements) was demonstrated[16]. Carrierless phase-retrieving coherent measurements in single mode fibers using a multimode scrambler were also demonstrated using a two-dimensional photodiode array[17]. It should be stressed that phase retrieval techniques are important for various current and future applications of multi-mode or FMFs in imaging, sensing, delivery of high-power coherent beams, nonlinear fiber optics, neuromorphic photonics, medical applications, and others. In this work, our focus is not on a particular application, but on the development of an advanced signal processing algorithm that can be applied across these fields.

Growing interest in FMF stimulates the demand for efficient beam characterization algorithms at the fiber output. The simplest approach is to measure the $M^2$ factor of the beam[18], which, however, considers only the beam divergence. A full description of the beam includes a characterization of the amplitudes and phases of the waveguide eigenmodes. This problem is known as mode decomposition (MD).

Several approaches based on use of a reference beam such as digital holography[19,20] and multi-plane light conversion[21,22] have been proposed. However, implementation of these methods requires a coherent radiation source on the receiver side that limits their applicability.

A number of methods without a reference beam have been proposed to solve the MD problem. Numerical computing-based MD methods include the classical Gerchberg–Saxton technique[23], line-search[24], and stochastic parallel gradient descent[25]. Methods include iterative procedures such as gradient descent or genetic algorithms. Although iterative methods show a high accuracy and

a performance that makes it possible to decompose several times a second, they are still sensitive to the initial value and can become stuck at a local minima.

Non-iterative methods for MD include using the fractional Fourier system[26] or machine learning methods[27–31]. Several neural networks architectures have been proposed either to enhance performance of iterative methods by guessing the initial mode weights distribution[30] or for direct application for the MD problem[27,29] in FMFs. MD methods using neural networks outperform iterative methods in decomposition speed; however, they require high-performance computers, a large amount of memory, and a long time for training the neural networks. In addition, they cope poorly when a fiber supports more than five modes.

The iterative algorithm based on stochastic parallel gradient descent presented in ref. [32] makes it possible to decompose nine images per second in a three-mode fiber. A hybrid genetic global optimization algorithm[33] allows only one decomposition per 150 s for six-mode fiber in noiseless case, although it does not get stuck at local minima. The fractional-Fourier method presented in ref. [26] makes it possible to solve the MD problem in up to 12-mode fibers. The combined CNN/gradient descent algorithm presented in ref. [30] allows up to 20 decompositions per second for three-mode fiber. The fully CNN-based approach presented in ref. [31] takes 30 ms per decomposition (33 frames per second) for three-mode and five-mode fibers, with added noise up to 20 dB SNR. Overall, previously published results show MD in FMFs with up to six modes, achieve an MD time above tens of milliseconds. In previously published papers where the effect of noise on MD was studied, the SNR ratio was limited to 20 dB.

However, previously published results on intensity-only are very limited in terms of the number of modes and are characterized by relatively long processing time.

In this work, we propose a non-iterative algorithmic method for MD without using a reference beam. The method is based on dividing the inherently non-linear MD problem into two parts: a cumbersome linear part and a simple non-linear part. Such approach allows not only to drastically decrease the decomposition time, but also shows a substantial increase in the number of modes that can be resolved in the noiseless problem. The MD method presented in this paper contributes to the rapidly developing field of MD in FMFs with a promising new opportunity. Namely, we would like to stress a significant progress in the time performance: the proposed method allows us to decompose up to 100,000 frames per second for three-, five-, and eight-mode fibers, which makes it—to the best of our knowledge—the fastest intensity-only MD method presented to date. In a model experiment without adding noise, we show the applicability for MD in a 27-mode fiber. The stability of the algorithm to the noise level in the input signal is not inferior to previously published results for FMFs.

## Results

**Method description**. Transverse distribution of an electric field in a fiber can be represented as a linear combination of eigenmodes $\Psi_k$:

$$E(x, y) = \sum_k C_k \Psi_k(x, y). \tag{1}$$

Here $C_k = A_k \exp(i\varphi_k)$ are complex coefficients representing amplitudes and relative phases of eigenmodes.

One can measure intensity experimentally, which is

$$I(x, y) = \left\langle |E(x, y)|^2 \right\rangle. \tag{2}$$

Adding a constant phase shift to every phase coefficient will not affect the output intensity, so without loss of generality, we assume $\varphi_1 = 0$.

The problem of MD is to determine the coefficients $C_k$ using the intensity distribution at the output of the fiber.

For a fiber that supports $N$ eigenmodes, it is necessary to determine $N$ amplitudes and $N-1$ phases (as we assume $\varphi_1 = 0$), a total of $2N-1$ coefficients.

If the intensity distribution is captured by a camera or an array of photodiodes, the obtained image can be used to recover the coefficients $C_k$. Consider an input image consisting of $M \times M$ pixels. Then Eq. (2) can be written as a system of $M^2$ equations:

$$I^{(m)} = \sum_k \sum_j C_k C_j^* \Psi_k^{(m)} \Psi_j^{(m)}, \quad m = 1..M^2. \quad (3)$$

Denote

$$z_n = \frac{C_k C_j^* + C_j C_k^*}{2} \quad k,j = 1..N, \quad n = 1..N(N+1)/2. \quad (4)$$

The one-to-one correspondence between the index $n$ and the indices $k$ and $j$ is shown in the matrix below. We numerate the vector $\mathbf{z}$ along the main diagonal first, and then along the columns of the lower triangular matrix $\mathbf{Z}$:

$$\mathbf{Z} = \begin{pmatrix} z_1 & 0 & \cdots & & 0 \\ z_{N+1} & z_2 & 0 & & \\ z_{N+2} & z_{2N} & z_3 & \ddots & \vdots \\ \vdots & z_{2N+1} & \vdots & \ddots & 0 \\ z_{2N-1} & \cdots & & z_{N(N+1)/2} & z_N \end{pmatrix} \quad (5)$$

$$\equiv \begin{pmatrix} C_1 C_1^* & 0 & \cdots & 0 \\ \frac{C_2 C_1^* + C_1 C_2^*}{2} & C_2 C_2^* & 0 & \vdots \\ \vdots & & \ddots & 0 \\ \frac{C_N C_1^* + C_1 C_N^*}{2} & \frac{C_N C_2^* + C_2 C_N^*}{2} & \cdots & C_N C_N^* \end{pmatrix}.$$

Note that

$$\begin{cases} z_n = A_k A_j = A_k^2, & k = j, \quad n = 1..N \\ z_n = A_k A_j \cos(\varphi_k - \varphi_j), & k \neq j, \quad n = N+1..N(N+1)/2 \end{cases}. \quad (6)$$

Now Eq. (3) can be written in the matrix form

$$\mathbf{I} = \mathbf{T}z. \quad (7)$$

Here the matrix $\mathbf{T}$ is of size $M^2$ by $N(N+1)/2$ and the $m$th row of this matrix includes pairwise products of $\Psi_k^{(m)} \Psi_j^{(m)}$:

It should be noted that if Eq. (7) does not have exact solutions (for example, due to a presence of a noise component in the experimentally obtained vector $\mathbf{I}$), several methods such as LMSE and approximate message passing[34] can be used to infer the vector $\mathbf{z}$. In this work, we use pseudoinverse Moore–Penrose matrix and find the vector $\mathbf{z}$ using Eq. (8).

We can rewrite vector $\mathbf{z}$ in matrix form (see Eq. (5)):

$$\mathbf{Z} = \begin{pmatrix} z_1 & 0 & \cdots & & 0 \\ z_{N+1} & z_2 & 0 & & \\ z_{N+2} & z_{2N} & z_3 & \ddots & \vdots \\ \vdots & z_{2N+1} & \vdots & \ddots & 0 \\ z_{2N-1} & \cdots & & \cdots & z_N \end{pmatrix}$$

$$\equiv \begin{pmatrix} Z_{1,1} & 0 & \cdots & & 0 \\ Z_{2,1} & Z_{2,2} & 0 & & \\ Z_{3,1} & Z_{3,2} & Z_{3,3} & \ddots & \vdots \\ \vdots & \vdots & \vdots & \ddots & 0 \\ Z_{N,1} & Z_{N,2} & & \cdots & Z_{N,N} \end{pmatrix}$$

or, considering Eq. (6)

$$\mathbf{Z} = \begin{pmatrix} A_1^2 & 0 & \cdots & & 0 \\ A_2 A_1 \cos(\varphi_2) & A_2^2 & 0 & & \\ A_3 A_1 \cos(\varphi_3) & A_3 A_2 \cos(\varphi_3 - \varphi_2) & \ddots & & \vdots \\ \vdots & & \vdots & \ddots & 0 \\ A_N A_1 \cos(\varphi_N) & \cdots & A_N A_k \cos(\varphi_N - \varphi_k) & \cdots & A_N^2 \end{pmatrix}. \quad (9)$$

Now one can easily derive all amplitude coefficients:

$$A_n = \sqrt{Z_{n,n}}. \quad (10)$$

Knowing $A_n$, we can determine cosine values for all the phase coefficients using the first column of matrix $\mathbf{Z}$:

$$\cos(\varphi_k) = \frac{Z_{k,1}}{A_1 A_k}. \quad (11)$$

Note that the replacement of all $C_k$, $k = 1...N$ with their complex conjugates $C_k^*$, $k = 1...N$ leads to the same intensity distribution. Therefore, we only need to determine the phase coefficients up to complex conjugation. To do that, we

$$\mathrm{T} = \begin{pmatrix} \Psi_1^{(1)} \Psi_1^{(1)} & \cdots & \Psi_N^{(1)} \Psi_N^{(1)} & 2\Psi_1^{(1)} \Psi_2^{(1)} & \cdots & 2\Psi_1^{(1)} \Psi_N^{(1)} & 2\Psi_2^{(1)} \Psi_3^{(1)} & \cdots & 2\Psi_2^{(1)} \Psi_N^{(1)} & \cdots & 2\Psi_{N-1}^{(1)} \Psi_N^{(1)} \\ \vdots & \vdots & \vdots & \vdots & & \vdots & \vdots & & \vdots & & \vdots \\ \Psi_1^{(m)} \Psi_1^{(m)} & \cdots & \Psi_N^{(m)} \Psi_N^{(m)} & 2\Psi_1^{(m)} \Psi_2^{(m)} & \cdots & 2\Psi_1^{(m)} \Psi_N^{(m)} & 2\Psi_2^{(m)} \Psi_3^{(m)} & \cdots & 2\Psi_2^{(m)} \Psi_N^{(m)} & \cdots & 2\Psi_{N-1}^{(m)} \Psi_N^{(m)} \\ \vdots & \vdots & \vdots & \vdots & & \vdots & \vdots & & \vdots & & \vdots \\ \Psi_1^{(M^2)} \Psi_1^{(M^2)} & \cdots & \Psi_N^{(M^2)} \Psi_N^{(M^2)} & 2\Psi_1^{(M^2)} \Psi_2^{(M^2)} & \cdots & 2\Psi_1^{(M^2)} \Psi_N^{(M^2)} & 2\Psi_2^{(M^2)} \Psi_3^{(M^2)} & \cdots & 2\Psi_2^{(M^2)} \Psi_N^{(M^2)} & \cdots & 2\Psi_{N-1}^{(M^2)} \Psi_N^{(M^2)} \end{pmatrix}.$$

Here the upper index $(m)$ corresponds to the $m$th pixel of the image of an eigenmode.

Equation (7) can be easily solved

$$z = \mathbf{T}^{-1}\mathbf{I}. \quad (8)$$

Here $\mathbf{T}^{-1}$ is a pseudoinverse (Moore–Penrose inverse) matrix and $\mathbf{z}$ is a vector determined in Eqs. (5) and (6).

simply choose

$$\varphi_2 = +\mathrm{acos}\left(\frac{Z_{2,1}}{A_1 A_2}\right), \quad \varphi_2 \in [0, \pi]. \quad (12)$$

We should note that, without loss of generality, we assume $\varphi_1 = 0$.

Signs of other phase coefficients for $k > 2$ can be chosen depending on $Z_{k,2}$:

$$\varphi_k = \begin{cases} +\mathrm{acos}\left(\frac{Z_{k,1}}{A_1 A_k}\right), & \text{if} \quad \frac{Z_{k,2}}{A_k A_2} = \cos\left(+\mathrm{acos}\left(\frac{Z_{k,1}}{A_1 A_k}\right) - \varphi_2\right) \\ -\mathrm{acos}\left(\frac{Z_{k,1}}{A_1 A_k}\right), & \text{if} \quad \frac{Z_{k,2}}{A_k A_2} = \cos\left(-\mathrm{acos}\left(\frac{Z_{k,1}}{A_1 A_k}\right) - \varphi_2\right) \end{cases}.$$

We choose the sign that corresponds to the minimum discrepancy between $\cos(\varphi_k^{\pm} - \varphi_2)$ and $Z_{k,2}/(A_k A_2)$. If there is no noise in the input image, then Eq. (7) has an exact solution and the discrepancy is exactly zero for either $\varphi_k^+$ or $\varphi_k^-$.

Thus, we recover all amplitude and phase coefficients.

We implemented the algorithm and tested its performance against the number of eigenmodes, the resolution of input image, and the noise of the input image.

At first, we choose external parameters of the problem, such as numerical aperture of the fiber, core radius, signal wavelength, and resolution of intensity image. Assuming the applicability of the weakly guiding approximation, we calculate eigen (LP) modes $\Psi_k(x, y)$ for this fiber at a given wavelength.

Then we calculate matrices $\mathbf{T}$ and its Moore–Penrose inverse $\mathbf{T}^{-1}$.

To measure the performance and accuracy of the algorithm, we generate a random set of amplitude and phase coefficients:

$$C_k^{\text{true}} = \left\{ A_k^{\text{true}}, \varphi_k^{\text{true}} \right\}, \quad A_k \in [0, 1], \quad \varphi_k \in [0, 2\pi].$$

We then calculate the true intensity distribution that corresponds to this set using Eq. (2):

$$I^{\text{true}} = \left| \sum_k C_k^{\text{true}} \Psi_k \right|^2.$$

Then we apply the decomposing algorithm to recover the coefficients $C_k^{\text{recov}} = \left\{ A_k^{\text{recov}}, \varphi_k^{\text{recov}} \right\}$ using only the intensity distribution $I^{\text{true}}$. After that, we calculate the intensity distribution that corresponds to the recovered coefficients:

$$I^{\text{recov}} = \left| \sum_k C_k^{\text{recov}} \Psi_k \cdot \right|^2.$$

To evaluate the performance and accuracy of the algorithm, we use the following metrics:

$$\varepsilon_{\text{A}} = \frac{\|A^{\text{recov}} - A^{\text{true}}\|}{\|A^{\text{true}}\|}, \quad \varepsilon_{\varphi} = \frac{\|\varphi^{\text{recov}} - \varphi^{\text{true}}\|}{\|\varphi^{\text{true}}\|},$$

$$\varepsilon_{\text{NET}} = \frac{\|C^{\text{recov}} - C^{\text{true}}\|}{\|C^{\text{true}}\|}, \quad \varepsilon_{\text{I}} = \frac{\|I^{\text{recov}} - I^{\text{true}}\|}{\|I^{\text{true}}\|}.$$

We also measure the decomposition time $T_{\text{calc}}$.

*MD with no added noise.* In the ideal case of zero noise in the input image, the algorithm shows the best performance in terms of accuracy and decomposition speed. There are examples of the MD for some of the external parameters as shown in Fig. 1.

With an increase in the number of modes, the accuracy of the recovering phase coefficients decreases faster than the accuracy for the amplitudes. Despite a relatively low amplitude error $\varepsilon_{\text{A}}$ for 29-mode fiber, the overall decomposition accuracy is low. So, the described method in theory works fine for the number of modes up to $N = 27$.

One can notice that phase errors are higher, in general, than amplitude errors, and we believe that this is caused by the calculation of amplitudes with an error and the subsequent calculation of the phase coefficients using those imprecise amplitude coefficients. More details on the maximum number of modes for this method and the nature of the restrictions are

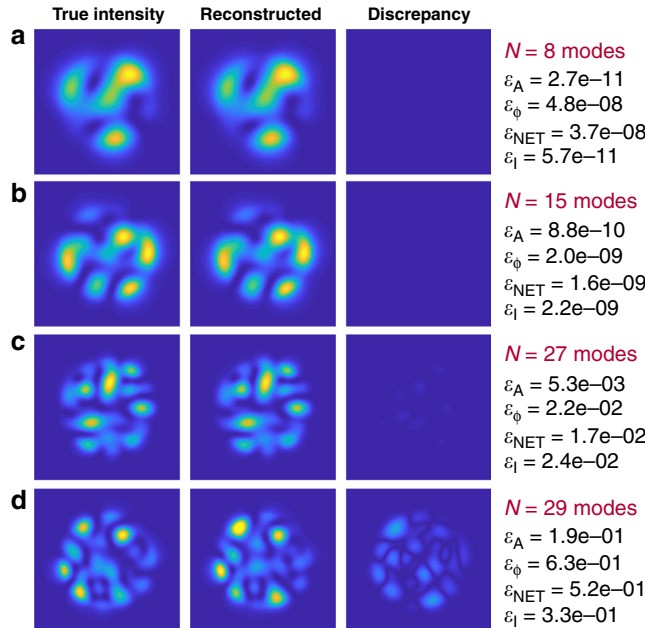

**Fig. 1 Examples of noiseless mode decomposition.** Mode decomposition for **a** 8-mode, **b** 14-mode, **c** 27-mode, **d** 29-mode fibers. Input image size of 100 × 100 pixels. Decomposition errors are shown on the right of each set of images. Here $\varepsilon_{\text{A}}$—amplitude error, $\varepsilon_{\varphi}$—phase error, $\varepsilon_{\text{NET}}$—total error, and $\varepsilon_{\text{I}}$—intensity error.

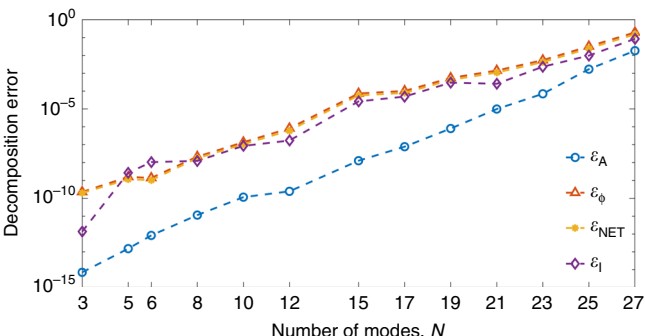

**Fig. 2 Noiseless decomposition errors.** Dependence of the amplitude, phase, net and intensity errors on the number of modes in the absence of noise. Here $\varepsilon_{\text{A}}$—amplitude error, $\varepsilon_{\varphi}$—phase error, $\varepsilon_{\text{NET}}$—total error, and $\varepsilon_{\text{I}}$—intensity error. Source data are provided as a Source Data file.

given in Supplementary materials (please refer to Supplementary Note 1).

There is always a tradeoff between the number of modes and the decomposition accuracy. Figure 2 shows how various error metrics depend on the number of modes.

Each point on the graph corresponds to an error value averaged over 10,000 decompositions.

*Time performance.* We investigated how the decomposition time depends on parameters of the problem. All tests have been performed on PC with CPU Intel 8700K. For a performance test, we performed 10,000 decompositions and averaged the calculation time. Figure 3 shows the dependence of mean decomposition time depending on the number of modes $N$ and the resolution of input image $M$.

One can see that the decomposition time is <10 μs for 3–8-mode fibers and the size of the processed image of 16 × 16. In order to evaluate performance of our method, we compared it

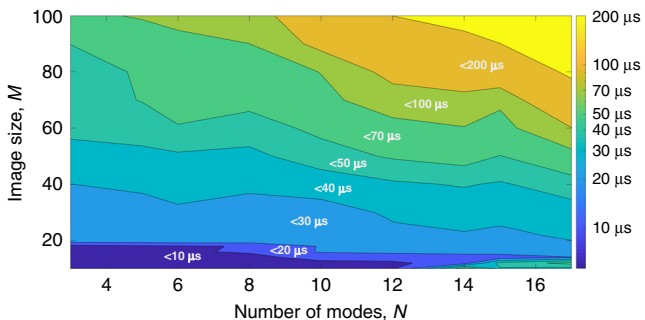

**Fig. 3 Time performance.** Mean decomposition time for different image sizes and number of modes. Decomposition time is color-coded. Source data are provided as a Source Data file.

**Table 1 Decomposition time (in seconds) using the deep learning algorithm.**

| M \ N | 3 | 5 | 8 |
|---|---|---|---|
| 32 | 3.13E−02 | 3.16E−02 | 4.20E−02 |
| 64 | 3.97E−02 | 4.29E−02 | 7.26E−02 |
| 128 | 7.44E−02 | 8.69E−02 | 2.17E−01 |

**Table 2 Decomposition time (in seconds) using the presented algorithm.**

| M \ N | 3 | 5 | 8 |
|---|---|---|---|
| 32 | 2.00E−05 | 2.12E−05 | 2.12E−05 |
| 64 | 3.03E−05 | 3.13E−05 | 3.14E−05 |
| 128 | 6.33E−05 | 6.58E−05 | 6.87E−05 |

with the fastest previously published method[31]. The direct comparison of the numerical algorithm developed here with the fastest MD method presented before is challenging, because the latter approach is based on GPU calculations. To compare these algorithms, we implemented the previously published deep learning MD method and tested its time performance using the same CPU as we were using for testing our MD algorithm. The computation time is averaged over 1000 decompositions. Decomposition time for the deep learning method depending on image size $M$ and the number of modes $N$ is presented in Table 1.

We tested our algorithm for the same parameters. The result is presented in Table 2.

The performance gain is summarized in Table 3.

Thus, we demonstrate that the proposed new method is more than 1000 times faster than the fastest method for MD presented before.

*MD with added noise.* For practical applications we need to investigate how the decomposition accuracy depends on the noise level in the input image. We apply an additive white Gaussian noise model to the intensity:

$$I^{\text{noisy}}(x, y) = \max[0, I^{\text{true}}(x, y) + N(0, \alpha) \cdot \max(I^{\text{true}})].$$

Here $N(0, \alpha)$—a normally distributed random matrix of the same size as $I^{\text{true}}$. Noise factor $\alpha$ determines the variance of the noise. Max[$0, x$] function is applied to avoid negative values of intensity.

**Table 3 The comparative performance gain (times).**

| M \ N | 3 | 5 | 8 |
|---|---|---|---|
| 32 | 1565 | 1491 | 1981 |
| 64 | 1310 | 1371 | 2312 |
| 128 | 1175 | 1321 | 3159 |

For each pair $\alpha$, $M$ we performed decompositions for 10,000 random sets $C_k^{\text{set}}$ with added noise. Then error metrics were averaged over these 10,000 decompositions.

It turns out that for decomposition accuracy there is always a trade-off between number of modes and noise level. Figure 4 shows how decomposition errors depend on the noise level for various numbers of modes for the fixed image size of $100 \times 100$ pixels.

Figures 5 and 6 show how the amplitude error and the net error depend on the noise factor and resolution of the input image. The vertical axis corresponds to the signal-to-noise-ratio in the input image, and the horizontal axis corresponds to the image size $M$. The color shows the decomposition error in decibels.

Figures 5 and 6 show that the method can be applicable for MD with a net error of $10^{-1}$ at a noise level of $10^{-2}$ for 3-mode fiber and at a noise level of $10^{-3}$ for 5-mode fiber correspondingly.

For 6–8-mode fibers, the noise requirements are further enhanced. For higher-mode fibers, the noise level needed to reliably determine the amplitudes and phases of the modes is too low for the method to be applied in practice. However, we anticipate that more accurate noise consideration can improve MD accuracy. Due to non-negativity of the intensity noise, its distribution is non-Gaussian, which induces additional errors in the mode weight distribution. For more information on the error distribution please see the Supplementary materials (please refer to Supplementary Note 2).

As mentioned above, phase errors and consequently net errors increase faster than the amplitude errors with SNR decreasing.

*Experimental verification.* We also performed an experimental verification of our MD algorithm. The experimental setup is shown in Fig. 7.

Laser light with a wavelength of 650 nm passed through a silica fiber with a numerical aperture of 0.14 and a core diameter of 5.3 μm. The normalized frequency is $V = 3.59$, and the fiber supports three eigenmodes at this wavelength: $LP_{01}$, $LP_{11}^{\text{o}}$ and $LP_{11}^{\text{e}}$. We used a 4-f imaging system to expose the near field on a CCD camera. The quantization of the camera was 10 bits, corresponding to a total of 1024 different intensity levels. The focal lengths of the lenses are $f_1 = 4.51$ mm and $f_2 = 300$ mm, which corresponds to a magnification of about 66.5. We also used a polarizer to select only one polarization component. By rotating the polarizer by 90°, we were able to decompose both polarizations. The decomposition speed was much higher than the speed of our camera (60 Hz), so, initially, we recorded a series of images in both polarizations and then applied the decomposition algorithm to test its speed and accuracy.

The resulting image was cropped to the size of $128 \times 128$ pixels. Then the background intensity level was subtracted from each image. This background was non-zero even when the main lighting in the lab was turned off and it was mainly caused by the beam light scattered by elements of the setup as well as a residual lighting, e.g. from the PC screen. This resulted in a pedestal observed in each measured image. Figure 8 shows (a) a measured image, (b) the pedestal outside the main beam, and (c) a histogram of intensity distribution in the pedestal. The mean

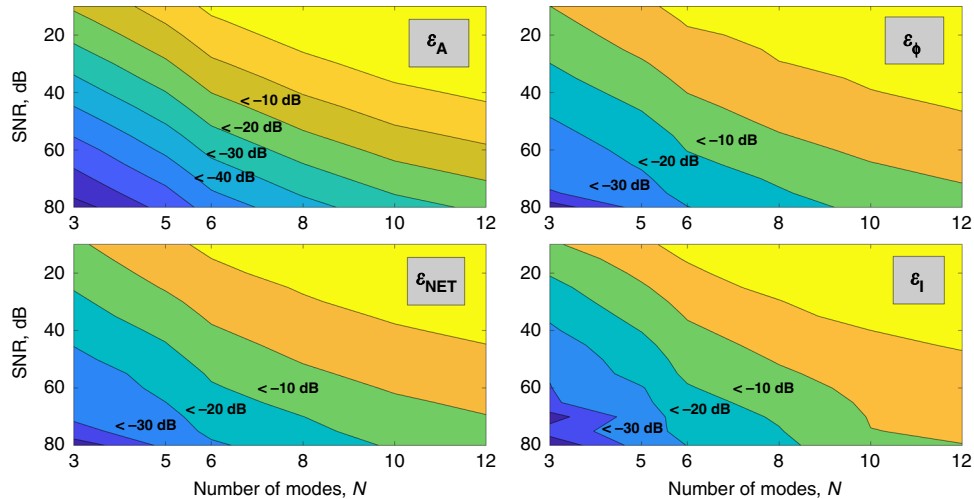

**Fig. 4 Decomposition errors depending on noise level and number of modes.** Decomposition errors depending on SNR and number of modes for 100 × 100 input image: **a** amplitude error, **b** phase error, **c** net error, and **d** intensity error. Horizontal axes correspond to the number of modes and vertical axes correspond to SNR in dB. Decomposition error values are color-coded, regions with an error value less than a certain value are labeled on the graphs. Here $\varepsilon_A$—amplitude error, $\varepsilon_\varphi$—phase error, $\varepsilon_{NET}$—total error, and $\varepsilon_I$—intensity error. Source data are provided as a Source Data file.

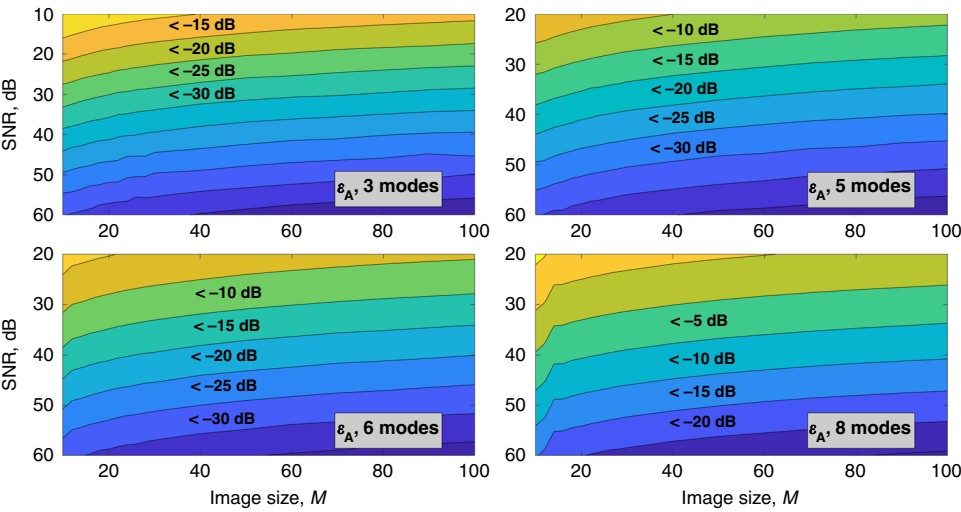

**Fig. 5 Dependence of the amplitude decomposition error on noise level and image size.** Amplitude decomposition error $\varepsilon_A$ for **a** 3-mode, **b** 5-mode, **c** 6-mode, and **d** 8-mode fibers depending on image size and noise level in the input image. Horizontal axes correspond to the image size in pixels, vertical axes correspond to SNR value in dB. Amplitude decomposition error values are color-coded, regions with an error value less than a certain value are labeled on the graphs. Source data are provided as a Source Data file.

intensity value of the pedestal was subtracted from each image and then the MD was performed. Noise level was calculated as a standard deviation of the pedestal intensity. The measured SNR value is 23 dB.

All images were processed using our algorithm and then images were reconstructed using the recovered weights and phases. The accuracy of MD was checked by calculating the correlation[24] between the captured and the reconstructed images:

$$\mathrm{Corr} = \left| \frac{\iint \Delta I_m(\mathbf{r})\Delta I_r(\mathbf{r})\mathrm{d}^2\mathbf{r}}{\sqrt{\iint \Delta I_m(\mathbf{r})^2\mathrm{d}^2\mathbf{r} \cdot \iint \Delta I_r(\mathbf{r})^2\mathrm{d}^2\mathbf{r}}} \right|. \quad (13)$$

Here $I_m$, $I_r$ are measured and reconstructed intensity distributions, $\Delta I_j(\mathbf{r}) = I_j(\mathbf{r}) - \bar{I}_j$, $j = $ m, r, and Corr is the correlation factor between captured and reconstructed images.

Figure 9 shows experimentally measured images in near field and their correlation with the reconstructed images using our MD algorithm.

The algorithm shows quite high overall decomposition accuracy and excellent time performance. Mean correlation is above 0.99 over all captured and decomposed images.

The proposed method can be easily expanded for the MD of both polarizations—it is enough to install a polarization beam splitter instead of a polarizer and to measure the intensity distribution for both polarization states.

## Discussion

We have proposed a new technique for the phase retrieval in FMFs using intensity-only measurements. We have demonstrated an excellent time performance of the method for FMFs: the decomposition time is as small as 10 μs, which is 1000 times better than the decomposition time using convolutional neural

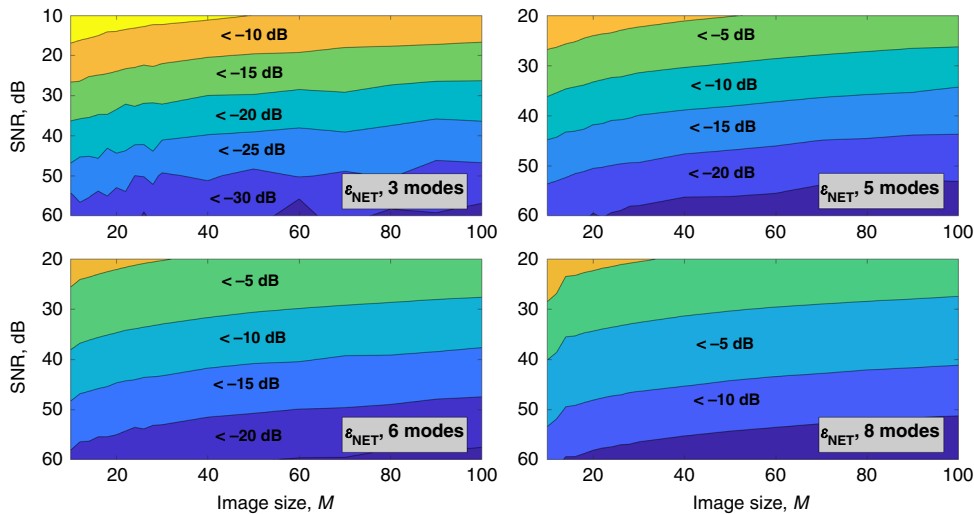

**Fig. 6 Dependence of the net decomposition error on noise level and image size.** Net decomposition error $\varepsilon_{NET}$ for **a** 3-mode, **b** 5-mode, **c** 6-mode, and **d** 8-mode fibers depending on image size and noise level in the input image. Horizontal axes correspond to image size in pixels, vertical axes correspond to SNR value in dB. Net decomposition error values are color-coded, regions with an error value less than a certain value are labeled on the graphs. Source data are provided as a Source Data file.

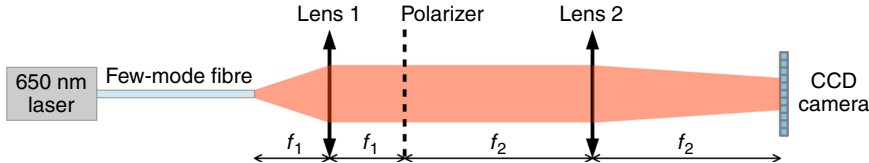

**Fig. 7 Experimental set up.** We used a 4-f imaging system to expose the near field of the few-mode optical fiber on a CCD camera for mode decomposition. A polarizer was used to select one polarization.

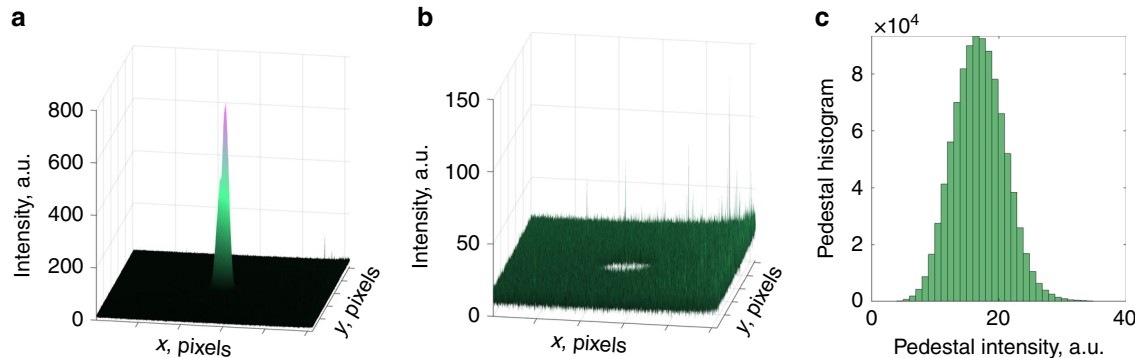

**Fig. 8 Accounting for intensity pedestal and calculating noise from the experimental data.** Figure shows **a** measured intensity profile, **b** pedestal intensity distribution, and **c** histogram of the pedestal. Mean decomposition time is 63 µs for 128 × 128 images. The measured SNR value is 23 dB.

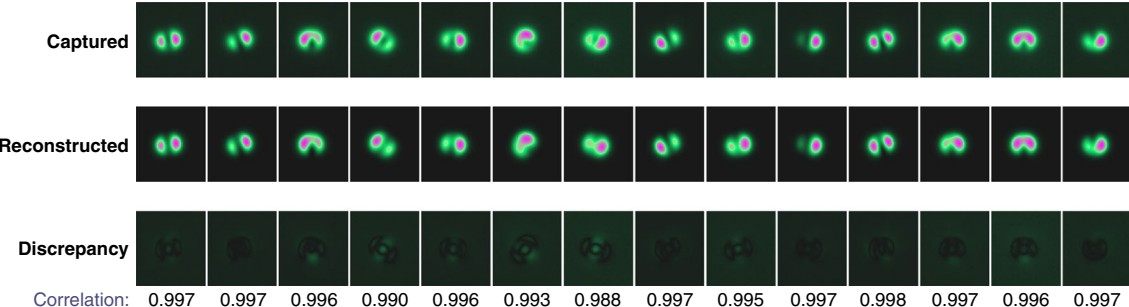

**Fig. 9 Experimental results.** Measured, reconstructed near field images, and their discrepancy and correlation.

networks. The proposed approach shows superior performance compared to other MD methods such as SGD and GA because it is non-iterative. It is worth mentioning that the proposed in this paper method does not require a reference beam in contrast to, for example, the digital holography method. Another important advantage of the technique is that it does not need an initial approximation, to which iterative methods are sensitive and may become stuck at a local optimum. Although the methods based on the convolutional neural networks also do not require any initial assumption, they are currently applicable to fibers with a number of eigenmodes of only up to 12 for a noiseless MD.

We have demonstrated that our method is capable for MD of no-added-noise images for number of modes of up to $N \approx 27$, which is the best achieved result as compared to the current optimization approaches and CNN methods. The number of modes that our technique can currently recover is limited by the structure of matrix $\mathbf{T}$, which becomes ill-conditioned with the increase in number of modes (see Supplementary naterials).

At present, we have successfully demonstrated for the 3-mode and 5-mode fibers that our method is capable of recovering both amplitude and phase coefficients when the noise factor $\alpha$ is about $10^{-2}$ and $10^{-3}$, correspondingly. For the eight-mode fiber, only intensity coefficients can be recovered when the noise factor $\alpha$ is lower than $10^{-4}$. Decomposition of noisy images requires a higher signal-to-noise ratio for higher-order mode fibers.

In general, an increase in the resolution of the input image results in the improved decomposition accuracy; at the same time, the decomposition time elongates. To increase the accuracy further, more sophisticated methods can be applied for inferring vector $\mathbf{Z}$, e.g. a generalized message passing algorithm. Inference of vector $\mathbf{Z}$ using probabilistic approach can benefit from incorporating an appropriate noise model in the inference problem.

It is worth mentioning that our new mathematical algorithm outperforms state-of-the-art deep learning-based methods, illustrating the importance of fresh mathematical ideas in this field.

In spite of remaining technical challenges, we believe that the reported here efficient and computationally simple MD techniques together with recent advances[35] in real-time evaluation of multi-mode fibers' transfer matrix using single-ended channel estimation are important steps towards the development of future cost-efficient receivers for spatial-division multiplexing systems.

## Methods
The fiber used in the experiments is a standard commercially available Corning HI1060 fiber. The laser used in the experiments is a standard telecom semiconductor fault locator with a central wavelength of 650 nm. The camera for capturing intensity patterns is Ximea mq013rg-e2. All numerical simulations are performed on a PC with CPU Intel 8700k. LP mode calculation, MD algorithm and all performance tests were written using MATLAB R2019b.

## Data availability
The data that supports the findings of this study is available from the corresponding author upon reasonable request. Source data are provided with this paper.

## Code availability
Code for implementing the mode decomposition algorithm is available from the corresponding author upon reasonable request.

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

## Acknowledgements

This work was supported by the EPSRC Program grant TRANSNET (EP/R035342/1). E.S.M. acknowledges support of the of H2020 MSCA COFUND Program MULTIPLY, V.V.D. and S.K.T. acknowledges support by the Russian Science Foundation (Grant no. 17-72-30006). We would like to thank David Saad for useful discussions.

## Author contributions

S.K.T. initiated the study. E.S.M. designed and conducted the numerical modeling. E.S.M. conceived the experiment and carried it out. S.K.T. and E.S.M. guided the theoretical investigations. E.S.M. and V.V.D. analyzed the data. E.S.M., V.V.D., and S.K.T. wrote the paper.

## Competing interests

The authors declare no competing interests.
