## [Peer Review File · Nature Communications]

Reviewers' comments:

Reviewer #2 (Remarks to the Author):

This paper presents a new mode-decomposition (MD) algorithm that is applicable to multimode optical fibers, based on the detected intensity of the near-field in the transverse plane at the fiber output, which is claimed to be both more accurate and faster than the state-of-the-art MD methods. There is currently a considerable research and applicative interest in the use of multimode optical fibers, for example, in optical communications to increase channel capacity via space-division multiplexing, in biomedical imaging for improving image resolution, in super continuum sources and in fiber lasers, to scale up the output energy. The present work appears to be focused on optical communications applications, which typically involve few mode fibers and use MIMO techniques for compensating mode coupling and modal dispersion. In these applications, the main bottleneck is provided by the complexity of the MIMO digital data processing, which currently prevents its use in real-time demultiplexing of the transmitted data.

The present algorithm is based on a direct matrix inversion of a linear system of equations relating the measured intensities at each pixel of the output image with products of pairs of the complex mode amplitudes. Being a linear problem, the method is solved by standard matrix inversion and regularization (being generally an ill-posed problem in the presence of noise) techniques. The recovery of the mode amplitudes then becomes a straightforward final nonlinear step. To my knowledge, this work provides an original approach to solving the MD problem, which has potential to give a progress in the field of information transfer by multimode fibers. However, there are several issues that remain to be addressed by its Authors, before the impact of this paper can be fully assessed. Specifically:

1. A key issue is the computational speed improvement of the presented method with respect to other methods. In the abstract, it is written that the method is "more than 10 times faster than the state of the art". The minimum MD time is of the order of tens of microseconds. But later in the main text, on line 172, it is written that the method is 1000 times faster than the fastest method reported before. Which value is true? Besides, the quoted computation times are obtained with a PC based on Intel 8700K CPU. What CPU was used in other papers for their performance tests? Any comparison only makes sense if made by using the same machine. Therefore the Authors should implement other state of the art methods with their own PC, and make fair speed comparisons. In particular, using the same number of modes N and same input image resolution M .
2. In the experimental section, on line 225, it is written that the MD was done with "excellent time performance", but no time performance value is provided.
3. Again on the computation speed issue: on line 251 it is written that their method "being implemented on FPGA, can cope with telecom symbol rate". How is that justified? Which symbol rate are the Authors thinking about? Please be quantitative and justify how one can improve by, say, at least 5 orders of magnitude the speed (from less than 0.1 MHz up to tens of GHz) by means of a dedicated hardware.
4. The references in the manuscript are almost totally messed up. It makes it very difficult for a reader to guess which paper the Authors have in mind when they make their citations. For example, ref.(1) on imaging should probably be ref.(18), ref.(2) on microwave photonics should probably be ref.(21), ref.(3) on sensing should be ref.(12)... then refs.(4-8) have nothing to do with solitons and self-cleaning (I guess the Authors mean refs.11, 17, 19, 22 here)... and so on.
5. On line 5, "that both the" should be "that use both the"
6. On line 56, SPDG is not defined.
7. About the noise level, the present method does not appear to be superior to other methods: on line 65 it is written that currently the SNR ratio is limited to 20 dB, and in line 197 it is concluded that the present method can be applicable to a MD with the same 20 dB SNR for a 3 mode fiber. Besides, there is a typo

there since it is written that the SNR is 20 dB for a 3 mode fiber and 30 dB for a 3 mode fiber (again). What do the Author mean?

8. On line 100, by "string" do the Authors mean "row"?

9. On line 180 it is written "the figures below": which figures? The next figure below is figure 4 which does not depend on the resolution of image M but the number of modes N. Again on line 186, "next figure", what figure do they mean?

10. In the conclusion it is written on line 244 that for a 3 mode fiber the maximum noise level is 30 dB of SNR but as discussed on point 7, the noise level was reported at 20 dB.

11. In the experimental verification, in addition to the accuracy expressed in term of correlation, the level of noise in the detected image should be reported.

Reviewer #3 (Remarks to the Author):

Coherent optical receivers capture the full field of the optical signal by interference means. The recovery of both the phase and the amplitude of the signal allows to effectively undo chromatic dispersion and mode dispersion in the electronic domain. On the contrarily, direct detection schemes only retrieve intensity information and therefore cannot compensate transmission impairments. For many emerging applications ranging from communications to microendoscopy, the modal decomposition of a multimode beam based on intensity measurements is of great importance. In this paper the authors report on the theoretical and experimental demonstration of a mode decomposition algorithm with a decomposition time of tens of microseconds based only on intensity measurements. The problem is written as a matrix problem: $I = Tz$, where I intensity, T product of modes matrix, T complex coefficients matrix. Then it can be found that the pseudoinverse matrix T^{-1} – as the solution is not unique. The proposed method is very elegant and allows full field reconstruction from intensity measurements only.

I find the experimental work incomplete and I have several comments below that in my opinion do not support publication in Nature Communication of the manuscript in its current form.

In their experimental demonstration, a fiber supporting three modes at 650nm is employed, with no mention of the time required for this case neither the number of pixels, the figure of merit is based in correlation values. A proper formulation of this experiment accounting for polarization and a larger number of appears to be highly essential.

Numerically, the authors present a validation of the method for 100*100 pixels. Showing that without noise up to 27 modes can be decomposed confidently, the error level $\sim 10^{-1}$, decomposition time of 10-200 microseconds. In the presence of noise, the number of modes considered is up to 8, and the number of pixels and SNR are also explored. However, these numerical results are not supported experimentally. Moreover, the limitations of the method in regards to number of modes, and other transmission impairments (modal dispersion) should be experimentally studied in a publication

In my opinion, the introduction is not clearly written and comprehensive. I find the introduction confusing and doesn't clearly state the relevance of the work with respect to the state-of-the-art.

As described by the authors, the main idea that inspired this work is the field of SDM multiplexing in optical communications. However, the authors don't clearly reference recent advances in the field to describe the context of their work. Moreover, it would be interesting to compare the presented method with other the results presented in ref [4] which demonstrates field reconstruction after transmission through a few mode fiber.

The paper is on a very interesting subject. However, in my opinion, the experimental results are not convincing enough to merit publication in Nature Communications.

Dear reviewers,

Thank you for useful and constructive comments. We revised the manuscript accordingly, in particular, we extended experimental part and put focus on the novel aspects of the algorithm and performance. Below we provide the detailed list containing the particular queries and our corresponding line-by-line responses right below each comment.

Reviewer 1

1. A key issue is the computational speed improvement of the presented method with respect to other methods. In the abstract, it is written that the method is “more than 10 times faster than the state of the art”. The minimum MD time is of the order of tens of microseconds. But later in the main text, on line 172, it is written that the method is 1000 times faster than the fastest method reported before. Which value is true? Besides, the quoted computation times are obtained with a PC based on Intel 8700K CPU. What CPU was used in other papers for their performance tests? Any comparison only makes sense if made by using the same machine. Therefore the Authors should implement other state of the art methods with their own PC, and make fair speed comparisons. In particular, using the same number of modes N and same input image resolution M.

The sentence that contained “10 times speed gain” is a typo, the text is corrected.

We agree with the comment and in the revised paper we provide details how we came to the number 1000. Indeed, the direct comparison of the numerical algorithm developed here with the fastest MD method presented before is challenging, because the latter approach is based on GPU calculations. To compare the algorithms themselves, we implemented the previously published deep learning MD method and tested its time performance using the same CPU as we were using for testing our MD algorithm. The computation time is averaged over 1000 decompositions. Decomposition time in seconds for the deep learning method depending on image size M and number of modes N is presented in the Table 1:

M \ N	3	5	8
32	3.13E-02	3.16E-02	4.20E-02
64	3.97E-02	4.29E-02	7.26E-02
128	7.44E-02	8.69E-02	2.17E-01

Table 1. Decomposition time (in seconds) using the deep learning algorithm.

We tested our algorithm for the same parameters. The result is presented in the Table 2:

M \ N	3	5	8
32	2.00E-05	2.12E-05	2.12E-05
64	3.03E-05	3.13E-05	3.14E-05
128	6.33E-05	6.58E-05	6.87E-05

Table 2. Decomposition time (in seconds) using the presented algorithm.

The performance gain is summarized in the Table 3:

M \ N	3	5	8
32	1565	1491	1981
64	1310	1371	2312
128	1175	1321	3159

Table 3. The comparative performance gain (times).

Thus, we demonstrated that the proposed new method is, indeed, more than 1000 times faster than the state-of-the-art advanced method for mode decomposition presented before.

This time performance comparison is added to the article.

2. In the experimental section, on line 225, it is written that the MD was done with “excellent time performance”, but no time performance value is provided.

Thank you for pointing that out, the experimental time performance is added to the article.

3. Again on the computation speed issue: on line 251 it is written that their method “being implemented on FPGA, can cope with telecom symbol rate”. How is that justified? Which symbol rate are the Authors thinking about? Please be quantitative and justify how one can improve by, say, at least 5 orders of magnitude the speed (from less than 0.1 MHz up to tens of GHz) by means of a dedicated hardware.

We completely agree, and we have changed the statement in a revised version. The idea is that the whole algorithm consists of matrix multiplication and several simple operations of addition or subtraction. Recently published results on FPGA Acceleration of Matrix Multiplication [https://www.xilinx.com/support/documentation/application_notes/xapp1332-neural-networks.pdf] uses pipelining which drastically increases matrix multiplication performance compared with general-purpose processing units. However, we agree that to make such a strong statement as applicability for telecommunication symbol rate, experimental justification should be provided. Therefore, this statement is excluded from the revised version. We also rewrote the paper to refocus it from the telecom application claims that require substantial additional work, to the detailed description of the results that might have applications in various fields.

4. The references in the manuscript are almost totally messed up. It makes it very difficult for a reader to guess which paper the Authors have in mind when they make their citations. For example, ref.(1) on imaging should probably be ref.(18), ref.(2) on microwave photonics should probably be ref.(21), ref.(3) on sensing should be ref.(12)... then refs.(4-8) have nothing to do with solitons and self-cleaning (I guess the Authors mean refs.11, 17, 19, 22 here)... and so on.

We apologize for the confusion. The whole reference list is now corrected and double-checked.

5. On line 5, “that both the” should be “that use both the”

Corrected.

6. On line 56, SPDG is not defined.

Abbreviation is replaced by a full description of the method in the article: SPGD = Stochastic parallel gradient descent.

7. About the noise level, the present method does not appear to be superior to other methods: on line 65 it is written that currently the SNR ratio is limited to 20 dB, and in line 197 it is concluded that the present method can be applicable to a MD with the same 20 dB SNR for a 3 mode fiber. Besides, there is a typo there since it is written that the SNR is 20 dB for a 3 mode fiber and 30 dB for a 3 mode fiber (again). What do the Author mean?

There is a typo, and a noise level of 30 dB or lower allows to decompose amplitudes and phases in a 5-mode fiber. It is 20 dB for 3-mode fiber and 30 dB for 5-mode fiber. The correction is made in the text.

8. On line 100, by “string” do the Authors mean “row”?

The correction is made.

9. On line 180 it is written “the figures below”: which figures? The next figure below is figure 4 which does not depend on the resolution of image M but the number of modes N. Again on line 186, “next figure”, what figure do they mean?

The text of the article is adjusted to get rid of the ambiguity. The cited text corresponds to figures 5 and 6 and it is moved right below the figure 4. The text on line 186 corresponds to figure 4 and it is remained just above figure 4.

10. In the conclusion it is written on line 244 that for a 3 mode fiber the maximum noise level is 30 dB of SNR but as discussed on point 7, the noise level was reported at 20 dB.

The cited text is corrected, now it reads: “For three-mode and five-mode fiber, we have demonstrated that our method is capable of recovering both amplitude and phase coefficients when the noise factor α is around 10^{-2} and 10^{-3} correspondingly”

11. In the experimental verification, in addition to the accuracy expressed in term of correlation, the level of noise in the detected image should be reported.

Information on noise level is added to the article. Mean decomposition time is 63 μ s for 128x128 images. Description of noise evaluation procedure is added to the article. The measured SNR value amounted to 23 dB. There was a residual background lighting even when the main light in the lab was off. This was mainly caused by the beam light scattered by elements of the optical system as well as a residual background lighting e.g. from PC screen. This led to the fact that each measured image had a pedestal. Figure 1 shows (a) measured image, (b) the pedestal outside the main beam and (c) a histogram of intensity distribution in the pedestal. Mean intensity value of the pedestal was subtracted from each image and then the mode decomposition was performed. Noise level was calculated as a standard deviation of the intensity in the pedestal.

Fig. 1. Measured intensity profile (a), pedestal intensity distribution (b) and the histogram of the pedestal (c). Mean decomposition time is 63 μ s for 128x128 images.

Reviewer 2

1. In their experimental demonstration, a fiber supporting three modes at 650nm is employed, with no mention of the time required for this case neither the number of pixels, the figure of merit is based in correlation values. A proper formulation of this experiment accounting for polarization and a larger number of appears to be highly essential.

We agree. The information concerning experimental image size as well as SNR and decomposition time is added to the article. As for polarization, there is a polarizer film in the focal plane of the 4-f imaging system that allows only one polarization component to pass through. By rotating the polarizer by 90 degrees, we could decompose both polarizations. The decomposition speed was much higher than the speed of our camera (60 Hz), so at first, we recorded series of images in both polarizations and then applied decomposition algorithm to test its speed and accuracy.

2. Numerically, the authors present a validation of the method for 100*100 pixels. Showing that without noise up to 27 modes can be decomposed confidently, the error level $\sim 10^{-1}$, decomposition time of 10-200 microseconds. In the presence of noise, the number of modes considered is up to 8, and the number of pixels and SNR are also explored. However, these numerical results are not supported experimentally. Moreover, the limitations of the method in regards to number of modes, and other transmission impairments (modal dispersion) should be experimentally studied in a publication

Indeed, we found that theoretically, without noise, the proposed method can work for fibers supporting up to 27 modes. Of course, we did not claim that 27 modes can be implemented experimentally. This should be considered as an upper bound. Our study explores among other things, how real experimental implementation reduces this theoretical limit. In the real-life experiment, the decomposition error depends on SNR and number of modes significantly. We found that for 3-mode fiber SNR of 20 dB is needed to decompose the intensity and we experimentally demonstrated applicability of the proposed approach. For the 5-mode fiber, a level of about 30 dB is required, which we did not achieve in the experiment with our camera. This is partly due to the quantization capacity of the camera, which is 10 bits, total of 1024 quantization units. This means that the digitizing noise is higher than 1/1024, practically it is around 1/900. And partly with high noise of the pixels themselves, which amounted to 4 quantization units. Together, this gives about 23 dB SNR at peak intensity of 900 quantization units, which is not enough for decomposition in a 5-mode fiber. We would like to stress that even for 3-mode fiber this is an experimental demonstration that detecting only intensity we can resolve phases of the signal.

3. In my opinion, the introduction is not clearly written and comprehensive. I find the introduction confusing and doesn't clearly state the relevance of the work with respect to the state-of-the-art.

As described by the authors, the main idea that inspired this work is the field of SDM multiplexing in optical communications. However, the authors don't clearly reference recent advances in the field to describe the context of their work. Moreover, it would be interesting to compare the presented method with other the results presented in ref [4] which demonstrates field reconstruction after transmission through a few mode fiber.

We agree. In case we want to focus on telecom we have to do more detailed comparison. The cited paper is devoted directly to telecom, it involves experiment on transmission and BER measuring. We also plan to do this, and we will compare our future results with the cited paper (Ref. 4 in the original submission and Ref 13 in the revised version). This will require substantial efforts and time. However, our current work is a proof-of-principle for the new concept of recovery signal phases using intensity detection and novel algorithm that does not require deep learning and outperform the state-of-the-art by 1000 times. In the revised paper we stress on the obtained results and try to explain better their importance.

4. The paper is on a very interesting subject. However, in my opinion, the experimental results are not convincing enough to merit publication in Nature Communications.

We took into account the comments regarding the experiment, and substantially extended experimental part. We hope that these changes improve the presentation of the experimental part. We would like to stress that our focus here is to present a novel concept and support the proposed technique with the experimental results. We do believe that community might find various ways to build on this new concept. In the revised paper we provide new results that we believe are important for the future development of this direction, and for practical implementations.

In conclusion, we would like to thank again reviewers for useful and constructive comments that helped us to improve presentation of our results.

REVIEWERS' COMMENTS

Reviewer #2 (Remarks to the Author):

In their revised version, the Authors have properly addressed all issues raised in my first report. I recommend the present version of the manuscript for publication as is.

Reviewer #3 (Remarks to the Author):

The authors have improved the manuscript in this revised version. I think that the authors have addressed the points highlighted by the reviewers. I find that the work is significant and is applicable in a wide range of applications. However, in my opinion, the experimental results of the paper are insufficient to warrant publication in a broad context journal such as Nature Communications. I believe the paper deserves publication in a more specific journal.

The main experiment addresses mode decomposition in a 3 mode fiber. Although the authors tried to use a 5 mode fiber, the noise level was too high for the method to work.

I'm not convinced that the previous work relating to fast mode decomposition in FMF has comprehensively referenced, in particular methods using digital holography and multiplane light conversion devices have been omitted. The relevant works will surely be known to the authors and some of them may have been omitted by oversight. Moreover, even after the authors revisions, I still believe the introduction is missing important SDM works. In particular, only one general review paper on SDM is cited. It would be helpful to have a slightly more comprehensive review of the published works than at present.

The abstract is good but I would suggest changing line 17 "techniques, is more than 1000 times faster" for "orders of magnitude faster".

Figure 4,5 and 6 are difficult to follow they are congested. The contour plots are missing color-bars and in some cases labels indicating parameter being plotted. Also, the figure captions are very general and don't explain in detail what is plotted in each figure.

I suggest that the authors carefully read the manuscript as there are lines that will not be obvious to all readers. For example, Line 224 "In the figures below we examine..." - please replace below for the figure number.

In Line 232 the authors state that the method can be applicable for mode decomposition at a noise level of 10^{-2} for 3- mode fiber and at a noise level of 10^{-3} for 5-mode fiber correspondingly. However, this point should be expanded. What error values are considered in order to determine if the methods applicable vs non-applicable?

Reviewer #2 (Remarks to the Author):

In their revised version, the Authors have properly addressed all issues raised in my first report. I recommend the present version of the manuscript for publication as is.

We are happy to see that the revised version addressed all issues raised by the reviewer 2. We are grateful for comments that helped us to improve presentation of the results.

Reviewer #3 (Remarks to the Author):

The authors have improved the manuscript in this revised version. I think that the authors have addressed the points highlighted by the reviewers. I find that the work is significant and is applicable in a wide range of applications. However, in my opinion, the experimental results of the paper are insufficient to warrant publication in a broad context journal such as Nature Communications. I believe the paper deserves publication in a more specific journal.

The main experiment addresses mode decomposition in a 3 mode fiber. Although the authors tried to use a 5 mode fiber, the noise level was too high for the method to work.

1. I'm not convinced that the previous work relating to fast mode decomposition in FMF has comprehensively referenced, in particular methods using digital holography and multiplane light conversion devices have been omitted. The relevant works will surely be known to the authors and some of them may have been omitted by oversight. Moreover, even after the authors revisions, I still believe the introduction is missing important SDM works. In particular, only one general review paper on SDM is cited. It would be helpful to have a slightly more comprehensive review of the published works than at present.

Our experimental results confirmed the theoretical findings and we believe that the proposed algorithm can work across various applications and be of interest to readers of Nature Communications from different communities. In this paper, we focused on the MD methods that do not use reference beam. However, we agree with the reviewer that to show a broader context of the previous research the list of references can be extended, and it would be right to mention these methods in the introduction. We included references to papers on digital holography and multiplane light conversion and SDM in the introduction.

The abstract is good but I would suggest changing line 17 “techniques, is more than 1000 times faster” for “orders of magnitude faster”.

Agree. Correction is made.

2. Figure 4,5 and 6 are difficult to follow they are congested. The contour plots are missing color-bars and in some cases labels indicating parameter being plotted. Also, the figure captions are very general and don't explain in detail what is plotted in each figure.

We accept this point. Figure captions have been extended to better describe the parameters displayed on them. The color-bars were omitted for the purpose of compactness of the plots. At the same time, regions of high interest are directly labeled on these plots.

3. I suggest that the authors carefully read the manuscript as there are lines that will not be obvious to all readers. For example, Line 224 “In the figures below we examine...” - please replace below for the figure number.

Corrected. The whole manuscript has been double checked in order to get rid of any ambiguities.

4. In Line 232 the authors state that the method can be applicable for mode decomposition at a noise level of 10^{-2} for 3- mode fiber and at a noise level of 10^{-3} for 5-mode fiber correspondingly. However, this point should be expanded. What error values are considered in order to determine if the methods applicable vs non-applicable?

Thank you for pointing that out. The value of net decomposition error of 10^{-1} , that can be achieved at specified noise levels, is added to the text.

We would like to thank once more reviewers for constructive and useful comments that helped to improve presentation of our results.

Faithfully yours,

E.M. Manuylovich, V.V. Dvoyrin and S. K. Turitsyn